# Green Synthesis of Copper Oxide Nanoparticles Using *Camellia sinensis*: Anticancer Potential and Apoptotic Mechanism in HT-29 and MCF-7 Cells

**DOI:** 10.3390/ijms26157267

**Published:** 2025-07-27

**Authors:** Devanthiran Letchumanan, Suriani Ibrahim, Noor Hasima Nagoor, Norhafiza Mohd Arshad

**Affiliations:** 1Centre for Research in Biotechnology for Agriculture, Universiti Malaya, Kuala Lumpur 50603, Malaysia; devanthiran@um.edu.my (D.L.); hasima@um.edu.my (N.H.N.); 2Department of Primary Care Medicine, Faculty of Medicine, Universiti Malaya, Kuala Lumpur 50603, Malaysia; 3Department of Mechanical Engineering, Faculty of Engineering, Universiti Malaya, Kuala Lumpur 50603, Malaysia; 4Institute of Biological Sciences (Genetics and Molecular Biology), Faculty of Science, Universiti Malaya, Kuala Lumpur 50603, Malaysia

**Keywords:** copper oxide nanoparticles, *Camellia sinensis*, anticancer activity, apoptotic mechanism

## Abstract

The increasing prevalence of cancer necessitates the development of novel and effective therapeutic agents. This study evaluates the anticancer potential of biosynthesized copper oxide nanoparticles (CuO NPs) using *Camellia sinensis* extract against human colon and breast cancer cells. The CuO NPs were characterized using various techniques to confirm their structure, size, morphology, and functional groups. The average size of CuO NPs synthesized was 20–60 nm, with spherical shape. The cytotoxic effects of these CuO NPs reveal a dose-dependent reduction in cell viability with 50% inhibitory concentration (IC_50_) at 58.53 ± 0.13 and 53.95 ± 1.1 μg/mL, respectively. Further investigation into the mechanism of action was conducted using flow cytometry and apoptosis assays, which indicated that CuO NPs induced cell cycle arrest and apoptosis in cancer cells. Reactive oxygen species (ROS) generation, caspase activity assay, and comet assay were also performed to elucidate the underlying pathways, suggesting that oxidative stress and DNA damage play pivotal roles in the cytotoxicity observed. Overall, our findings demonstrate that biosynthesized CuO NPs exhibit notable anticancer activity against colon and breast cancer cells, with moderate selectivity over normal cells, highlighting their potential as a therapeutic agent due to their biocompatibility. However, further studies are required to validate their selectivity and safety profile.

## 1. Introduction

Cancer has become one of the most devastating diseases, with more than 14 million new cases each year, which are expected to continue rising. According to the statistics released by the World Health Organization (WHO), there were an estimated 20 million new cancer cases and 9.7 million deaths in 2022. With the advancement in nanotechnology, nanoparticles are gaining lots of significance due to their use in cancer therapy. Previous studies have reported that metal oxide nanoparticles can induce cytotoxicity in cancer cells with minimal side effects to the normal cells, possess antioxidant capacity that can decrease the rate of tumor cell progression, and induce generation of reactive oxygen species (ROS) oxidative stress, which causes DNA damage and increases death receptor expression [1,2].

Nanoparticles (NPs) are used in different fields, such as the chemical, food, electronic, and healthcare industries [3]. They can be synthesized by various methods, which can be categorized into bottom-up and top-down methods. In the bottom-up approach, atoms and molecules are utilized to synthesize complex nanostructures, while the top-down method involves the reduction of bulk particles to produce the desired nanostructure [4]. Although the synthesis process of NPs is designed to be straightforward and is based on a succinct procedure, the parameters for the synthesis must be carefully selected and controlled to obtain the desired NPs. Hence, careful consideration must be given to parameters such as temperature, time, type of precursor, pH, mixing ratio, and concentration prior to the synthesis of NPs.

Plant leaf extracts are rich in bioactive compounds such as alkaloids, amino acids, alcoholic compounds, and various chelating proteins, which are water-soluble and can easily reduce metal ions in a shorter time [5]. The advantages of using plant extracts over other biological organisms are that they are widely available, safe to handle, and contain different types of metabolites that function as reducing agents [6]. *Camellia sinensis*, also known as a tea plant, is a member of the theaceae family with evergreen trees or shrubs and is commonly used for beverages worldwide. Tea contains a high concentration of polyphenols and flavonoids, including the group known as catechins, which act as strong antioxidants, have metal-chelating properties, and possess cell protective effects.

Recently, the development of an efficient “green” chemistry method for synthesizing metal NPs has become a major focus for researchers. Synthesizing metal NPs using the green method is clean, nontoxic, cost-effective, and environmentally friendly. To date, CuO NPs have been synthesized using biological entities like bacteria [7], yeasts [8], fungi [9], and plants [10]. Among these, synthesis of CuO NPs using plants is well-known for the ease of scaling up, reduced biohazards, and stability [6,11]. The size, morphology, and stability of CuO NPs also can be easily optimized for medicinal and pharmaceutical usage using this green method [12,13].

Various studies have reported on the green synthesis of NPs. The natural product taxol, isolated from *Taxus baccata*, is a successful antitumor drug used in the clinical treatment of breast, ovarian, head and neck cancers, as well as lung cancer [14]. Vincristine from *Catharanthus roseus* is a good anticancer natural product used for the treatment of acute lymphocytic leukemia [15]. Apart from that, the plant *Hibiscus rosa-sinensis* extract has antitumor properties [16], *Murraya koenigii* possesses antioxidant and antitumor properties [17], *Moringa oleifera* has anti-proliferative properties and induces apoptosis in tumor cells and cytotoxicity against pancreatic cancer cells [18,19], and *Tamarindus indica* possesses anti-inflammatory and antibacterial properties [20].

In this study, we report on an eco-friendly green synthesis method for the fabrication of copper oxide nanoparticles (CuO NPs) with controlled size and morphology, mediated by the phytochemical reduction of copper ions using *Camellia sinensis* leaf extract. The biogenic synthesis of nanoparticles leverages the redox-active biomolecules within plant extracts, offering a sustainable alternative to conventional chemical methods while simultaneously enhancing nanoparticle biocompatibility for therapeutic applications. Although biogenic CuO NPs have demonstrated promising therapeutic properties, the molecular mechanisms underlying their anticancer activities, particularly those synthesized from tea leaf extracts, remain insufficiently characterized. Thus, this study focuses on the systematic evaluation of the cytotoxic, apoptotic, and molecular signaling effects of *C. sinensis*-mediated CuO NPs against human colorectal and breast carcinoma cell lines. We employ an integrated in vitro platform comprising cell viability, apoptosis quantification, reactive oxygen species (ROS) generation analysis, and genotoxic assessment. Through these mechanistic investigations, we aim to delineate the molecular events triggered by biosynthesized CuO NPs and validate their potential as novel, efficacious, and biocompatible nanotherapeutics for cancer treatment.

## 2. Results and Discussion

### 2.1. High-Resolution Transmission Electron Microscopy

To achieve precise characterization of the size, morphology, and structural integrity of the CuO nanoparticles, high-resolution transmission electron microscopy (HRTEM) was employed. Figure 1 shows the formation of spherical-shaped NPs. The average CuO NPs particle size ranges from 20 to 60 nm.

### 2.2. Elemental Analysis

EDX analysis of the synthesized CuO NPs using *C. sinensis* further confirms the reduction and formation of copper ions. The EDX results reflect the elements present in the synthesized CuO NPs according to their percentages. The elements copper, oxygen, and carbon, with the highest percentages, respectively, were observed in the CuO NPs, as shown in Figure 2. The presence of carbon elements is due to the tea extract, which consists primarily of organic compounds. This result can be compared to the study by Aardra et al., 2023, which identified the elements carbon (35.6%), oxygen (33.5), and copper (23.3%) [21].

### 2.3. Fourier-Transform Infrared Spectroscopy

The FTIR analysis was performed to identify phytochemicals that are responsible for the capping and stabilization of the synthesized NPs. From Figure 3, the spectrum shows a band at 834.21 cm^−1^, which refers to C–H bonding (aromatic and alkene) and the peak in the range of 450–530 cm^−1^, which corresponds to Cu–O metal oxygen vibration [22]. The band 1084.95 cm^−1^ refers to C–O bonding (alcohol), and 1370.16 cm^−1^ refers to C–H bonding (alkanes and alkyls). The band at 1571.46 cm^−1^ refers to C=C stretch (aromatic ring). The band at 3190.21 cm^−1^ refers to aliphatic and aromatic hydroxyl groups peak. The broadness of the peak is due to intermolecular hydrogen bonding among the hydroxyl group. This result is rather similar to the study by Dey and his colleagues [23].

### 2.4. Cytotoxicity of CuO NPs

Cell viability assays provide information about the reaction of cells to toxic substances, including their survival, death, and metabolism. As shown in Figure 4a in this study, MTT assay was utilized to evaluate the toxicity of CuO NPs towards RWPE-1 epithelial cell normal cells, HT-29 colon, and MCF-7 breast cancer cell lines. The commercial CuO NPs were used as a positive control to assess the cytotoxic potential of the test samples, as shown in Figure 4b. Each sample was incubated with the cells for 24 h at different concentrations to determine the cell viability. The results demonstrated that biosynthesized CuO NPs are cytotoxic in a dose-dependent manner against these cancer cell lines, where the percentage of viable cells decreased as the concentration of the CuO NPs increased. In addition, the normal RWPE-1 cell was employed to assess the selective cytotoxicity of CuO NPs between normal and cancerous cells. The IC_50_ value for RWPE-1 was found to be 83.88 ± 3.4 μg/mL. This result is in accordance with previous studies, which reported that the CuO NPs synthesized using *Pogestemon benghalensis* exhibit a half inhibitory effect on cell viability at 60.25 mg/L in human normal fibroblast HDF cell lines [24]. Apparently, CuO NPs has the lowest IC_50_ value against the MCF-7 cell lines, with a value of 53.95 ± 1.1 μg/mL in comparison to HT-29 cell lines with the value of 58.53 ± 0.13 μg/mL. Although cancer cells exhibited a slightly greater sensitivity to the biosynthesized CuO NPs compared to normal cells, the IC_50_ difference between tumor and normal cells was in the range of 1.4–1.6 fold. Thus, our cytotoxicity data demonstrate a modest degree of selectivity, as the IC_50_ values for cancer cell lines were only moderately lower than those for the normal epithelial cell line RWPE-1. This finding suggests that while CuO NPs exert more pronounced toxicity toward cancer cells, the selectivity margin is limited. The IC_50_ values for both normal and cancer cells were statistically significant, with a *p*-value < 0.05. The IC_50_ values for cells treated with commercial CuO NPs were 82.18 ± 0.15 μg/mL for RWPE cells, 22.46 ± 1.15 μg/mL for MCF-7 cells, and 51.46 ± 0.14 μg/mL HT-29 cells.

To further evaluate the mode of cell death, a dual-staining assay using fluorescent dyes-Cyto-Dye and propidium iodide (PI) was performed. Cyto-Dye selectively stains live cells with intact membranes, whereas PI penetrates and stains only cells with compromised membranes, typically indicative of late apoptosis or necrosis. This approach allows for clear differentiation between viable and non-viable cells based on membrane integrity and nuclear staining. As shown in Figure 5, treatment with the CuO NPs for 24 h resulted in a marked decrease in the percentage of viable cells in both HT-29 and MCF-7 cancer cell lines. Specifically, HT-29 cells exhibited a reduction in viability from 98.50% in untreated controls to 28.45% after treatment. Similarly, MCF-7 cells showed a decrease in viability from 97.50% to 35.75%. This significant loss of viable cells suggests that the compound exerts a strong cytotoxic effect on both cell lines, likely through induction of membrane damage and subsequent cell death. These findings are consistent with the initial MTT results and support the progression to further mechanistic studies to confirm apoptosis and elucidate the pathways involved.

In this study, copper oxide nanoparticles (CuO NPs) were synthesized utilizing *Camellia sinensis* extract as a biogenic reducing and stabilizing agent. A study reported by Zughaibi et al. (2022) on CuO NPs biosynthesized using pumpkin seed extract shows an IC_50_ value of 20 µg/mL when treated with MDA-MB-231 cells (breast cancer) [25]. While in another study of CuO NPs biosynthesized using *Abutilon indicum* leaf extract, an IC_50_ value of 14 μg/mL was reported when treated on the same breast cancer cell [26]. This indicates that the effect of anticancer properties varied when different plant extracts were administered. In another recent study, by Pillai et al. (2022), CuO NPs biosynthesized using *Pimenta dioica* leaf extract treated on DLD-1 cells (colorectal cancer) shows a higher IC_50_ value of 89.42 µg/mL to reduce 50% of the cells [27]. On the other hand, a study by Shinde et al. (2023), which biosynthesized CuO NPs using biowaste groundnut shell extract, shows an IC_50_ value of 42.66 μg/mL when treated on MCF-7 cells (breast cancer). This is very clear evidence that plants can be utilized as whole organisms in anticancer studies, especially when it comes to the synthesis of NPs [28]. NPs can also be explored through dose optimization, surface modification, and combination therapy through this biosynthesis.

### 2.5. CuO NPs Alter the Progression of Cancer Cell Growth

Cell cycle analysis was used to assess the effect of CuO NPs in preventing the proliferation of HT-29 and MCF-7 cancer cell lines. The cancer cells were treated for 24 h before fixing and staining with propidium iodide (PI) to examine the DNA contents using flow cytometry. The phase of cell-detecting diploid cells (G0/G1 phase) and tetraploids cells (G2/M phase), as well as the S phase located between the diploid and tetraploids states, are shown in Figure 6a.

The cell cycle profile for the HT-29 cell line in Figure 6(bi) showed an increase in cell population in the sub-G0/G1 phase from 15.5 ± 0.64% to 18.1 ± 1.48% after 24 h treatment. This sub-population of cells appears before the G0/G1 phase, which is also called the sub-G0/G1 (apoptosis) phase. At this stage, the cell populations are less stable due to subsequent DNA leakage in the cells. In the G0/G1 phase, the cell population decreased from 51.30± 1.70% to 43.40 ± 0.71% after 24 h treatment. HT-29 cell populations increased at the stage S phase and no major changes were observed in the G2 phase. On the other hand, in Figure 6(bii), it can be seen that the MCF-7 cell line exhibited a significant increase in the sub-G0/G1 phase from 16.4 ± 0.57% to 24.1 ± 1.56% after 24 h treatment. However, the cell population in the G0/G1 phase decreased from 57.0 ± 0.85% to 41.3 ± 0.42%, indicating the initiation of apoptosis. The S and G2 phases slightly increased during treatment after 24 h.

### 2.6. CuO NPs Induces Apoptosis in Cancer Cells: Annexin V-FITC Assay Test

The apoptotic effects of CuO NPs on the cancer cells were quantitatively investigated using the flow cytometry analysis and the Annexin V protocol: Annexin V-FITC assay. Propidium iodide is a standard flow cytometric viability probe used to distinguish viable from non-viable cells. The cell death induced by the CuO NPs follows a pathway from the lower-left quadrant to the upper-right quadrant (Annexin V^+^/PI^+^), which represents cells undergoing apoptosis.

Cells shift from the viable quadrant Q4 to the early apoptosis quadrant Q3 and eventually end up in the late apoptosis quadrant Q2. Furthermore, cells that undergo necrosis will shift from the viable quadrant Q4 to the late necrosis quadrant Q1, as shown in Figure 7. The CuO NPs induced apoptosis in HT-29 and MCF-7 cells by significantly increasing the population of cells undergoing early and late apoptosis after 24 h treatment, with no significant changes in the necrotic population.

Accurate identification of the cellular mode of death is critical in the rational design of therapeutic agents, as it directly influences treatment outcomes and patient prognosis. Apoptosis, a highly regulated form of programmed cell death, is an important type of cell death for maintaining cellular homeostasis, operating important processes involved in a wide range of functions, from the defense system to organ sculpting [29]. Cancer cells de-regulate the apoptosis machinery by activating oncogene and deactivating the tumor suppressor gene, allowing emission from cell death that leads to uncontrolled cell division, resistance towards drugs, and recurrence of tumors [30].

### 2.7. CuO NPs Induces Apoptosis in Cancer Cells: Activation of Caspase Pathways

To further elucidate the possible mechanisms of cancer cell death by CuO NPs, the activation and expression of caspase-3/7 proteins in cancer cells were analyzed. As shown in Figure 8, the results showed that caspase 3/7 protein expression is significantly higher in HT-29 and MCF-7 cells when treated with CuO NPs compared to the untreated cells. The caspase 3/7 protein concentration in HT-29 cells is 103.84 pg/mL and in MCF-7 cells it is 146.03 pg/mL. This shows that the caspase 3/7 protein expression level is higher when the cells are treated with CuO NPs, indicating the cells undergo cell death by apoptosis and CuO NPs may induce stronger apoptotic signaling in MCF-7 cells compared to HT-29 cells.

Caspase cascades can be triggered through two major branches, either through the cell surface TNF-superfamily death receptor and the pivotal adaptor protein Fas activated death domain (FADD), leading to activation of caspase-8 and, ultimately, caspase-3, or via the release of cytochrome C from mitochondria, resulting in caspase-9 activity and subsequent caspase-3 action. Caspase-3 represents a convergence point between the two routes, and in turn induces PARP cleavage, chromosomal DNA breaks, and, finally, the breakdown of the cell apoptotic bodies [31]. In this study, we use the human cleaved caspase 3/7 assay to evaluate CuO NP-treated cells. We found that the HT-29 colon and MCF-7 breast cancer cell lines exhibited a higher expression of caspase-3/7 proteins compared to the untreated control cells, indicating that the cells underwent cell death via apoptosis.

### 2.8. CuO NPs Damages the DNA of the Cancer Cells

Comet assay (single-cell gel electrophoresis) in an agarose gel matrix was used to study DNA fragmentation in cancer cells. When the comet assay was performed on both CuO NP-treated cancer cells, large and well-rounded comets were observed, while the untreated cells failed to display a comet-like appearance, as shown in Figure 9. The comet score for cancer cells treated with CuO NPs shows a significant number of nucleoids with larger comet tails, indicative of higher levels of DNA single-strand breaks.

Analysis of the level of DNA damage in the cancer cells was carried out based on the scoring (0, 1, 2, 3, and 4) from the lowest to highest level of DNA damage, respectively. A total of 100 cells were counted and scored for HT-29 and MCF-7 cells. Based on the results in Figure 9a, for HT-29 cells, 75% of untreated cells scored 0, decreasing to 2% when cells were treated with CuO NPs. Almost 45% of the cells treated with CuO NPs underwent severe DNA damage, with a score of 4. A similar trend was also observed in MCF-7 cells as the untreated cells with a score of 0 were 80% and after being treated with CuO NPs, the level of DNA damage with a score of 4 reached almost half of the cell population. This provides clear evidence that treatment of cancer cells with CuO NPs induced severe DNA damage and the cells underwent genotoxicity.

### 2.9. Mechanisms of Cancer Cell Death

Oxidative stress refers to the imbalance due to excess Reactive Oxygen Species (ROS) or oxidants over the capability of the cell to overcome and respond. Hence, oxidative stress leads to macromolecular damage and is involved in various disease effects, especially cancer. Generation of ROS, together with the release of pro-apoptotic proteins from the intermembrane space of mitochondria, triggers the activation of different modes of cell death. In this study, ROS was measured using Dichlorodihydrofluorescein-diacetate (H2DCFDA). The H2DCFDA passively enters the cell, where the acetyl group on H2DCFDA is cleaved by intracellular esterase and reacts with ROS to form the highly fluorescent compound. The fluorescence intensity is proportional to the ROS levels within the cell cytosol. The experimental values were expressed as a percentage of the fluorescence intensity relative to controls.

A higher level of fluorescence intensity shows a higher level of ROS generation in the cells. As shown in Figure 10, the ROS fluorescence intensity of HT-29 cells treated with CuO is 1523.34 Au, in comparison to 568.00 Au in the untreated cells, which is statistically significant (*p* < 0.05). However, the intensity is much higher in the MCF-7 treated cells, with a fluorescence intensity value of 1935.50 Au, compared to 513.3 Au in untreated cells, which is not statistically significant. These findings indicate that treatment with CuO NPs induces ROS generation in both cancer cell lines, thereby supporting the observed apoptotic and genotoxic effects. Consistent with this, the comet assay results showed a similar trend, with clear evidence of DNA double-strand breaks in the treated cells, as indicated by DNA scoring.

As reported by Chasara et al. (2023), apoptosis represents a key ROS-mediated mechanism, in which high levels of ROS trigger intracellular signaling cascades that lead to programmed cell death in cancer cells [32,33]. ROS exert their apoptotic effects by inducing DNA damage through oxidation of DNA bases, DNA strand breaks, and the formation of DNA adducts. The accumulation of DNA damage impairs cancer cells’ ability to replicate and repair their DNA, resulting in cell cycle arrest and, ultimately, cell death [32,34].

In many types of cancer, increased levels of reactive oxygen species (ROS) have been consistently observed in cancer cells compared to their normal counterparts. This elevation is attributed to various factors, including heightened metabolic rates, dysregulated cellular signaling, peroxisomal and mitochondrial dysfunction, oncogene activation, and the upregulation of ROS-generating enzymes such as oxidases, cyclooxygenases, lipoxygenases, and thymidine phosphorylases [32,35]. Therefore, the excessive ROS burden in cancer cells not only contributes to tumor progression and genomic instability but also renders these cells more susceptible to oxidative stress-induced cell death. Due to the duplex nature of ROS, methods to upregulate or downregulate ROS in cancer cells seem to hold potential for cancer treatment. 

## 3. Materials and Methods

### 3.1. Chemicals and Reagents

All materials were commercial reagent grade and used without further purification. All materials Cu(NO_3_)_2_(H_2_O)_x_, Folin Ciocalteu’s reagents, aluminum chloride colorimetric assay, gallic acid, and quercetin were purchased from the Sigma-Aldrich (St. Louis, MO, USA) and Merck Company (Darmstadt, Hesse, Germany). Deionized water was used for the preparation of all aqueous solutions. Fresh tea leaves were collected from Cameron Highlands, Pahang, in Peninsular Malaysia.

### 3.2. Green Synthesis of Copper Oxide Nanoparticles

Copper solutions with different molarities were prepared by diluting copper nitrate [Cu(NO_3_)_2_] in deionized water using a magnetic stirrer and stirring vigorously for 1 h at 350 rpm to obtain a well-mixed solution. Tea extract solutions were prepared by adding 200 mL of deionized water to 20 g of *C. sinensis* leaf for two hours at 60 °C. A dark yellow solution was created during the boiling procedure and allowed to cool at room temperature for further use. Copper oxide nanoparticles were prepared by adding 50 mL of tea extract and 450 mL of Cu(NO_3_)_2_ and stirring for 1 h at 60 °C. The color change in the aqueous solution from pale blue to pale yellow and finally to brown indicated the presence of copper oxide nanoparticles (CuO NPs). The CuO NPs were centrifuged for 15 min at 1500 rpm, washed with ethanol and deionized water to remove the impurities, and then dried in the oven at 40 °C overnight.

### 3.3. Material Characterization

#### 3.3.1. High-Resolution Transmission Electron Microscopy (HRTEM)

The size, shape, and distribution of CuO NPs were studied using a JEOL Ltd., JEM-2100F high-resolution transmission electron microscope (HRTEM) operating at 200 kV with a scale of 50 to 100 nm [36]. A total of 2 μL of each of the AgNP solutions was placed onto a carbon-coated copper grid and allowed to air dry. Size distribution analysis of the acquired TEM images was undertaken using the MIPAR–Image Analysis Software (version 3.8, MIPAR Software LLC, Columbus, OH, USA).

#### 3.3.2. Energy-Dispersive X-Ray Analysis (EDX)

The elemental composition of the air-dried, carbon-coated samples was examined using an energy-dispersive attachment on a transmission electron microscope (JEOL Ltd., JEM-2100F, Japan), under the following instrumental conditions: accelerating voltage of 15 keV and counting time of 100 s. The dispersal of X-rays from the CuO NPs was detected after exposing the CuO NPs to a high concentration of electrons [37].

#### 3.3.3. Fourier-Transform Infrared Spectroscopy

Fourier-transform infrared spectroscopy (FTIR) is useful for the identification of biomolecules involved in the capping and stabilization of CuO NPs. To understand and identify the surface chemistry of CuO NPs, FTIR was performed using a Thermo Nicolet 380 spectrometer Thermo Fisher Scientific Inc., Waltham, MA, USA, fitted with a SmartOrbit reflection accessory. The absorption of electromagnetic radiation was measured, with wavelengths falling within the mid-infrared region (4000–400 cm^−1^) [38]. The spectrum obtained from FTIR reveals information about the position of bands, which is related to the strength and nature of the bonds, and specific functional groups, which provide details about the molecular structures and interactions on the surface of NPs [39].

### 3.4. Cytotoxic Effects on Cancer

MCF-7 breast and HT-29 colon cancer cell lines were cultured in MEM-α and EMEM media, respectively, while normal RWPE-1 cells were cultured in KSFM media. Cells were allowed to grow as a monolayer at 37 °C in an incubator with 5% CO_2_ and 95% humidified atmosphere air. Cytotoxic activity of the CuO NPs was assessed using the (3-(4,5-dimethylthiazol-2-yl)-2,5-diphenyl-2H-tetrazolium bromide) (MTT) cell viability assay. Commercial CuO NPs were used as a positive control. Concentrations of 20, 40, 60, 80, and 100 μg/mL CuO NPs were treated on cells of 1 × 10^4^ plated in 96-well plates with a combination of 100 µL of media and cells. After incubation, the plates were measured at OD 560 nm using a microtiter plate reader, Tecan Group Ltd. (Männedorf, Zürich, Switzerland). In addition, live/dead assay was used to confirm the cell viability through fluorescence images captured using a Nikon Eclipse TS-100 fluorescence microscope (Nikon Corporation, Tokyo, Japan), with 100× magnification.

### 3.5. Apoptotic Effects on Cancer

Apoptosis activity was measured using a FITC Annexin V Apoptosis Detection Kit (BioLegend, San Diego, CA, USA) according to the manufacturer’s instructions. A total of 5 × 10^5^ cells were treated with CuO NPs at IC_50_ concentrations and were analyzed using MACSQuant^®^ Analyzer 10 (Miltenyi Biotec GmbH, Bergisch Gladbach, Germany) flow cytometry with MACSQuantify™ version 2.10 software. All the results were expressed in a scatter plot as total percentages of the cell population from four different quadrants representing different stages of cell death. On the other hand, caspase-3/7 activity was carried out to observe the initiation of apoptosis signal, using a Caspase 3/7 Assay kit, Abcam (Cambridge, UK). The caspase-3/7 activity in each sample was measured after 1-h incubation in the dark at 25 °C using GloMax®-Multi Jr 116 Single Tube Multimode Reader (Promega) (Promega Corporation, Madison, WI, USA).

### 3.6. Genotoxic Effects on Cancer Cells

To study the genotoxic effect of the CuO NPs, the comet assay technique recommended by Abcam (Cambridge, UK) was used. Cells were treated with IC_50_ concentrations and hydrogen peroxide was used as the positive control. After 24 h of treatment, cells were spread onto microscope slides precoated with normal agarose and subjected to the alkaline comet assay. Electrophoresis was performed at 4 °C under an electrical current of 400 mA (25 V) for 20 min. The comet images were analyzed using a fluorescence microscope 400× magnification, Eclipse 50i, Nikon (Nikon Corporation, Tokyo, Japan) and finally scored using Comet assay IV image analysis software (version 4.3, Perceptive Instruments Ltd., Suffolk, UK). The test parameter was referred to as the % DNA in the tail (tail intensity).

### 3.7. Mechanisms of Cell Death

Cell cycle analysis was used to demonstrate the influence of biosynthesized CuO NPs towards the growth of cancer cells. A total of 1.0 × 10^6^ cells were grown as a monolayer in a 6-well plate and treated with CuO NPs at the IC_50_ value and incubated for 24 h at a temperature of 37 °C in 5% CO_2_ and 95% humidified air. Cell pellets were obtained and stained with PI solution later analyzed using MACSQuant^®^ Analyzer 10 flow cytometry with MACSQuantify™ version 2.10 software. Mechanisms of cell death were studied in detail using a ROS detection kit purchased from Canvax Biotech, Spain (Canvax Biotech, Córdoba, Spain) A total of 1 × 10^4^ cells were cultured on 96-well plates and treated with CuO NPs at IC_50_ concentrations and hydrogen peroxide H_2_O_2_ was used as the positive control. Next, probe H_2_DCFDA dye was added into each well to stain the cells and the intensity was measured using the fluorescence microplate reader at Ex/Em = 485/530 nm.

### 3.8. Statistical Analysis

The experiments were conducted based on a completely randomized design (CRD) with three replications and the data were statistically analyzed using ANOVA to determine significant differences (defined as *p* ≤ 0.05) by using the SPSS software package (version 28, IBM Corp., Armonk, NY, USA). Fitting standard errors of the means (±SEM) were calculated for presentation of graphs.

## 4. Conclusions

In this study, the anticancer potential of biosynthesized CuO NPs and their molecular mechanisms against human breast and colon carcinoma cells were evaluated through a series of in vitro experiments. Overall, our findings demonstrate that CuO NPs biosynthesized using *Camellia sinensis* extract exhibit a potent anticancer activity against these two types of human carcinoma cells. The dose-dependent cytotoxicity, induction of apoptosis, and involvement of oxidative stress highlight the potential mechanisms through which these nanoparticles exert their effects. In addition, numerous biological experiments carried out have demonstrated the potential of this biosynthesized CuO NPs in various biomedical applications, especially in cancer treatments. The biosynthesis approach not only enhances the biocompatibility of CuO NPs but also aligns with sustainable and eco-friendly practices. While the findings support the anticancer potential of these nanoparticles, the limited differential toxicity and the absence of mechanistic data in normal cells warrant further investigation before conclusive claims of selectivity can be made. The integration of biogenic nanoparticle synthesis into cancer nanomedicine could pave the way for the development of novel, effective, and sustainable cancer treatments.

## Figures and Tables

**Figure 1 ijms-26-07267-f001:**
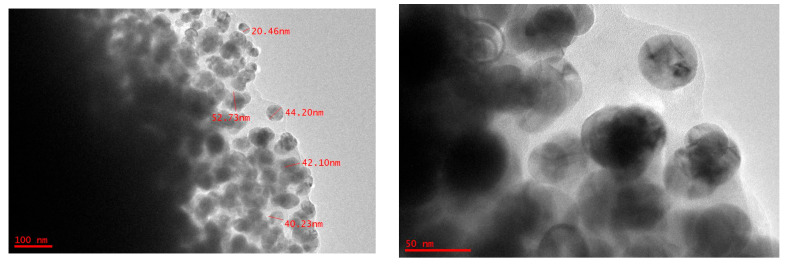
HRTEM images of CuO NPs. The images shows that the size of synthesized copper oxide particles ranges from 20 to 60 nm, with a spherical shape, as obtained from JEOL JEM-2100F (JEOL Ltd., Tokyo, Japan).

**Figure 2 ijms-26-07267-f002:**
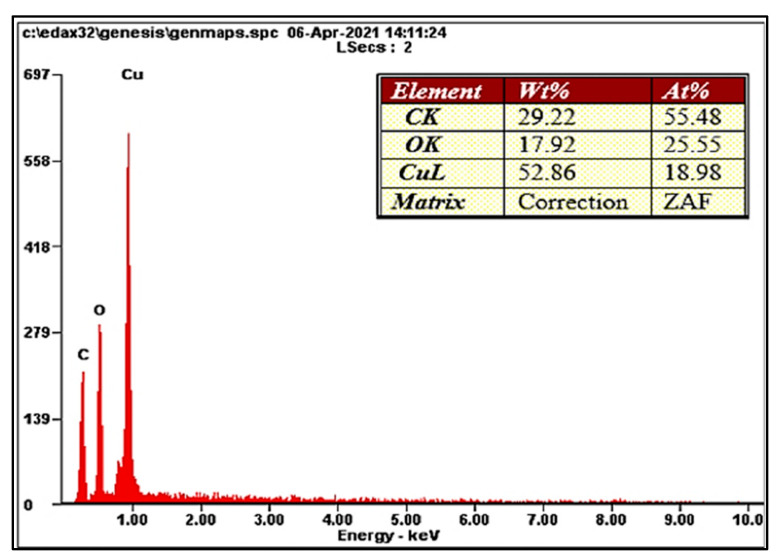
EDX spectrum of CuO NPs. EDX spectrum shows that the highest percentage of copper element can be obtained from the synthesized powder.

**Figure 3 ijms-26-07267-f003:**
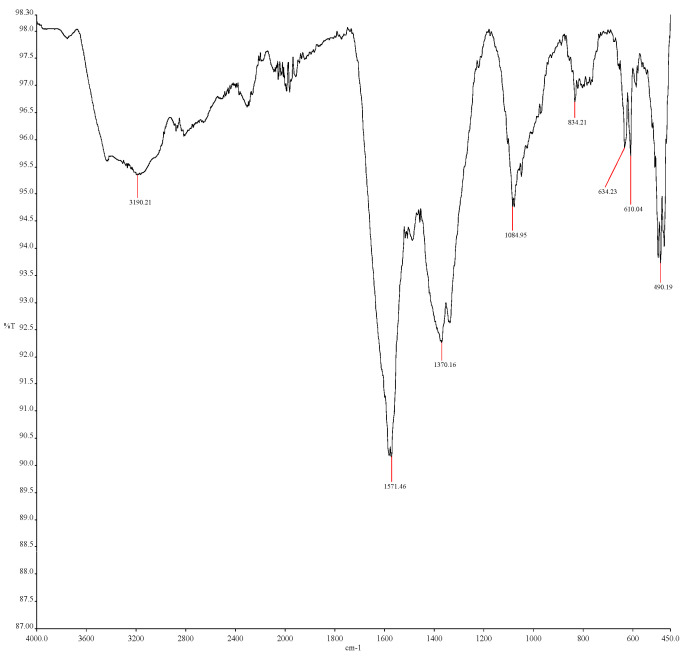
FTIR spectra of CuO NP shows the functional groups that consist of copper element and phytochemical elements on the copper oxide powder.

**Figure 4 ijms-26-07267-f004:**
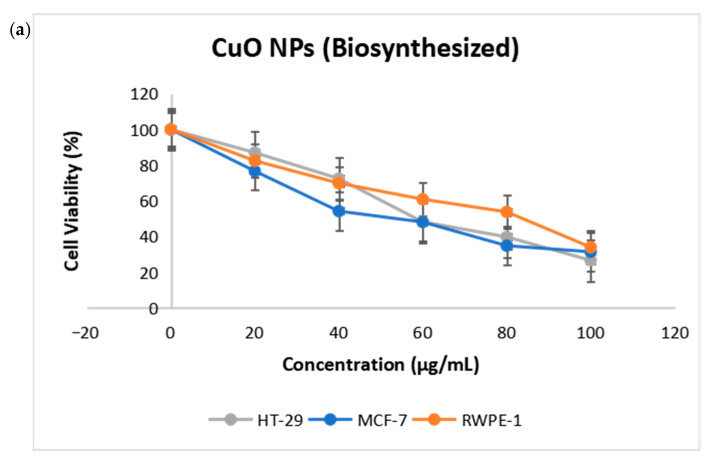
Cell viability (%) graph of HT-29, MCF-7, and RWPE cells treated with (**a**) CuO NPs (biosynthesized) and (**b**) CuO NPs (commercial) with 3 replicates with *p* < 0.05.

**Figure 5 ijms-26-07267-f005:**
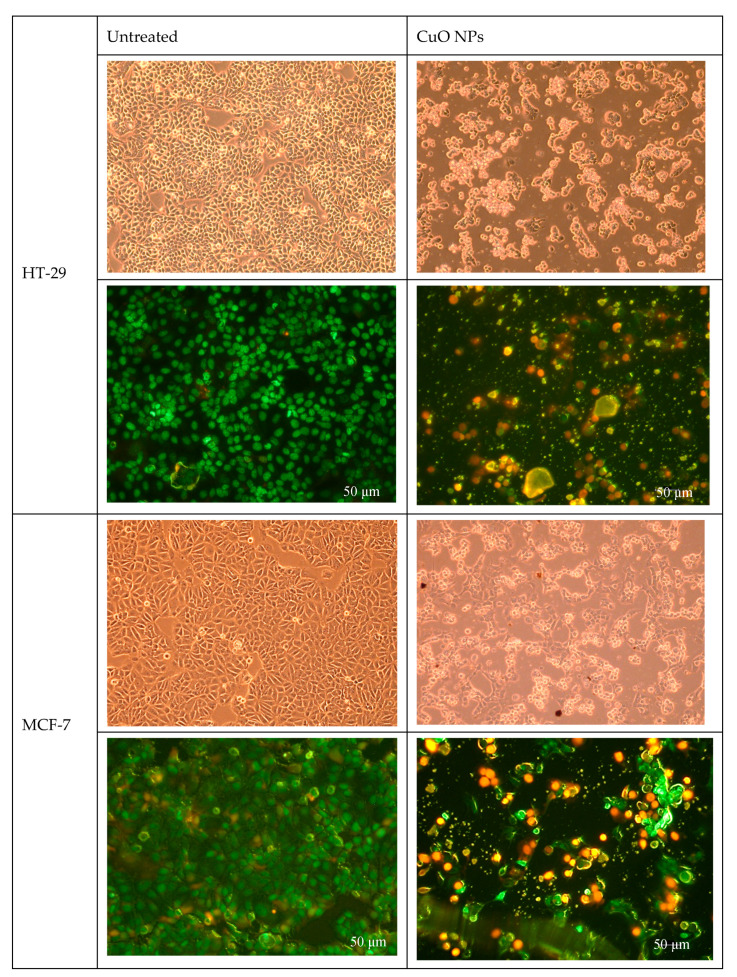
Microscopic and fluorescence images of HT-29 and MCF-7 after treatment with CuO NPs signifying early and late apoptosis. Cell morphological changes observed in comparison to the untreated cells using Nikon TS100 microscope (Nikon Corporation, Tokyo, Japan), with fluorescence (scale bar 50 μm; magnification 100×).

**Figure 6 ijms-26-07267-f006:**
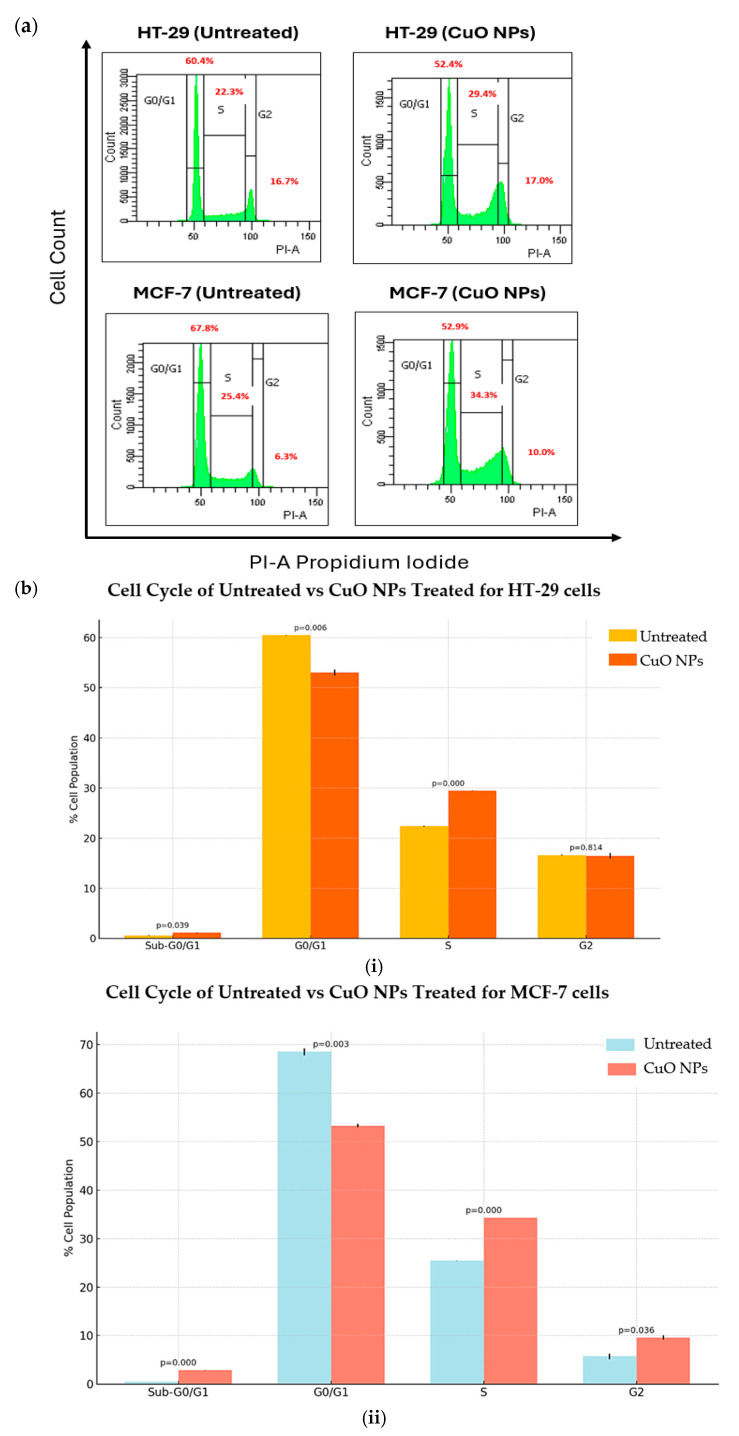
(**a**) Representative histogram of flow cytometry analysis and (**b**) Quantification of cell populations at different phases of cell cycles for (**i**) HT-29 and (**ii**) MCF-7 cells treated with CuO NPs compared to untreated control with significance difference (*p* < 0.05).

**Figure 7 ijms-26-07267-f007:**
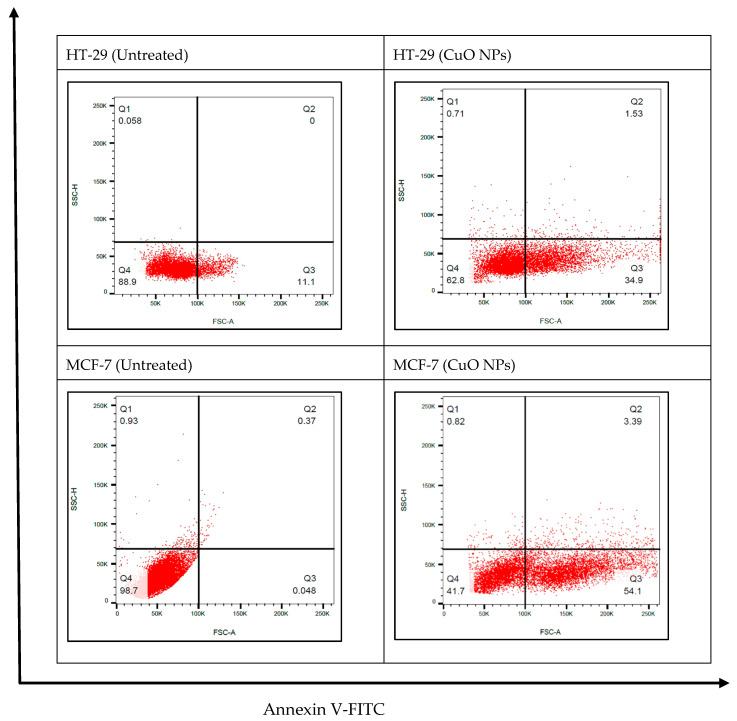
Annexin V-FITC assay result of HT-29 and MCF-7 cells treated with CuO NPs (*n* = 3 replicates).

**Figure 8 ijms-26-07267-f008:**
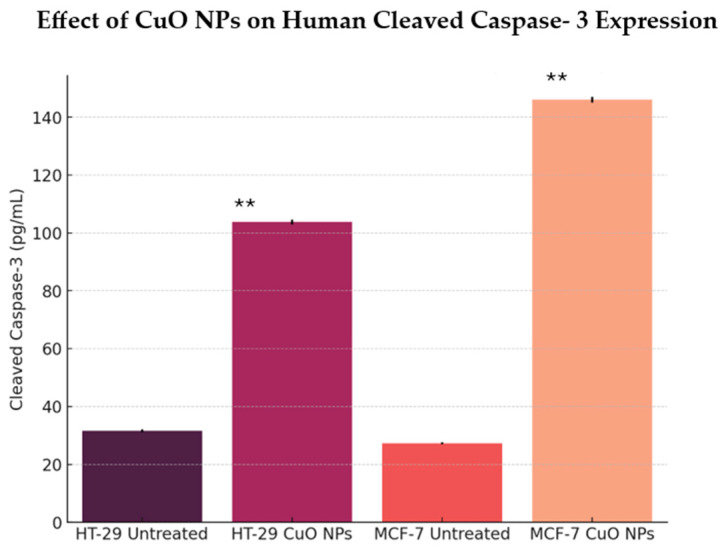
Quantification of caspase 3/7 protein expression of HT-29 and MCF-7 cells treated with CuO NPs (*n* = 3 replicates), with significance difference (*p* < 0.05 **).

**Figure 9 ijms-26-07267-f009:**
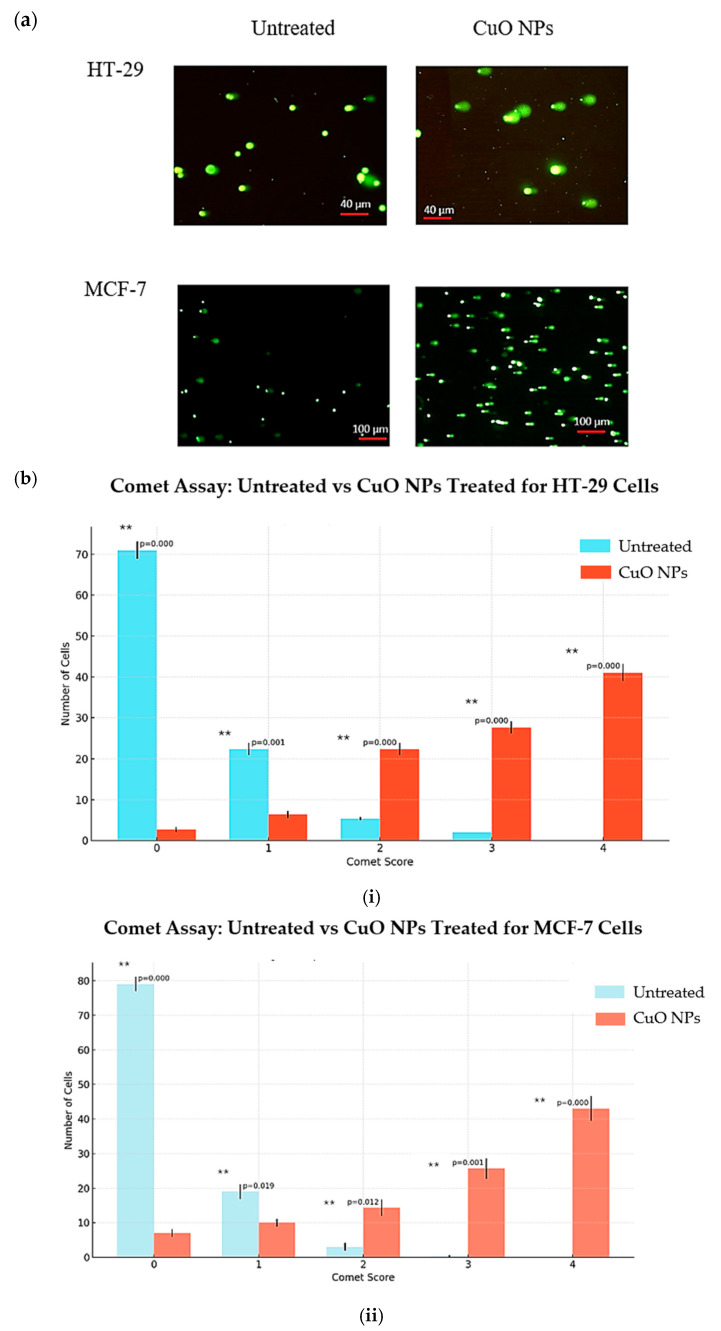
(**a**) Comet images captured using Nikon TS100 microscope with fluorescence; (**b**) Comet assay graph of control vs CuO NPs treated for (**i**) HT-29 cells and (**ii**) MCF-7 cells with significant difference (*p* < 0.05 **).

**Figure 10 ijms-26-07267-f010:**
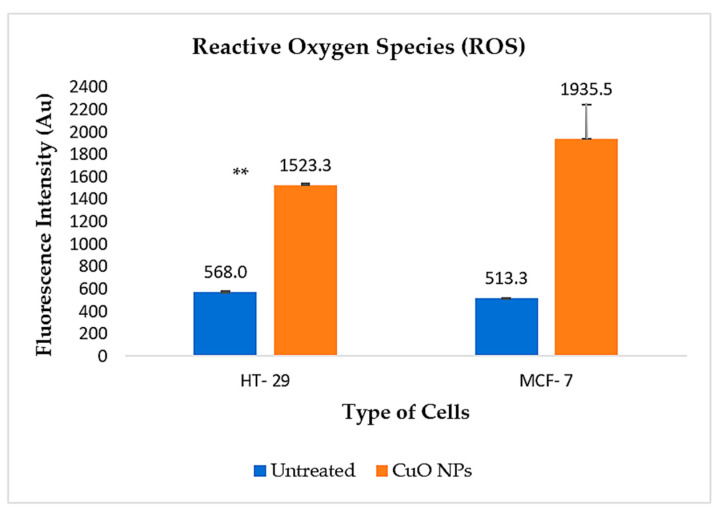
Quantification analysis of Reactive Oxygen Species (ROS) expression in HT-29 and MCF-7 cells, untreated and treated with CuO NPs with significant difference ** (*p* < 0.05).

## Data Availability

The original contributions presented in this study are included in the article. Further inquiries can be directed to the corresponding authors.

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
