# Peer review of "Green Synthesis of Copper Oxide Nanoparticles Using Camellia sinensis: Anticancer Potential and Apoptotic Mechanism in HT-29 and MCF-7 Cells"

_ijms, 2025, doi:10.3390/ijms26157267_

Round 1
Reviewer 1 Report
Comments and Suggestions for Authors
This article presents data on the study of the cytotoxic effect of copper oxide nanoparticles with tea plant extract. MCF-7 and HT cell lines were used as target cells. The methods are adequate.
Two significant shortcomings were found -
The materials and methods do not indicate how many experiments were repeated, what N.
It is necessary to supplement the materials and methods section with the following information - what was the number of repetitions in these experiments, were these biological repetitions, or repeated measurements? In figures 3 and 4, provide explanations regarding how the data are presented - median, mean, n.
In Figure 3b, the time of action of nanoparticles on cells should be indicated. In Figure 4b, very small errors of the mean are indicated, or are they not? How do you explain this?
Figure 5a - sign - "data from a flow cytometer are presented using the example of one experiment", Fig. 5b - the number n, in what the data are presented, how do you explain the extremely low variability?
Include a section on "Statistical data processing" in the materials and methods.
The second drawback is the lack of references to several very similar studies where these nanoparticles were also studied.
For example, the article doi: 10.1039/d0ra09924d describes a method for obtaining copper oxide nanoparticles using Camellia sinensis extract, the article doi: 10.7759/cureus.50220 examines the antioxidant activity of similar nanoparticles, and the article doi.org/10.1515/ntrev-2022-0081 examines the effect of similar nanoparticles on colorectal cancer cell line (HCT-116).
In addition, a question arose regarding the sterility of the nanoparticles used, was it assessed or was the synthesis carried out under sterile conditions?
It is also recommended to include information on the internalization of nanoparticles and their mechanisms of action in the "discussion" section.
Author Response
Dear reviewer,
Greetings.
Thank you for your constructive comments in improving the quality of our research work.
Here attached is the word document with your comments and responses added.
We hope this comments addressed will be satisfied.

Reviewer 2 Report
Comments and Suggestions for Authors
This study evaluates the anticancer potential of biosynthesized copper oxide nanoparticles (CuO NPs) using Camellia sinensis extract against human colon and breast cancer cells is some useful, but also there is some major problem need to be solve.(1) the author should explained why chose green Synthesis of Copper Oxide Nanoparticles;(2) The author should provied the data of green synthesis and not green systhesis;(3) the author should added more inforamtions such as size, PDI and stabilty data of nanoparticles;(4) The author should added more toxicity such as cell and animal of nanoparticle;(5) The author should discussion why is the apoptotic mechanism;(6) The author should added the postive drug control in this study.
Author Response
Dear reviewer,
Greetings.
Thanks for your comments and suggestions given to improve our research output.
We have managed to address most of the comments accordingly.
Hope the responses given is with satisfaction.

Reviewer 3 Report
Comments and Suggestions for Authors
The manuscript from Letchumanan et al. is an in vitro study investigating the anticancer effects of copper oxide nanoparticles biosynthesized by use of Camellia sinensis extract, mainly by use of flow cytometry assays and biochemical assays.
In the opinion of the reviewer the study cannot be published in the IJMS due to flaws in the experimental design, several major criticisms and also minor issues, present throughout the manuscript.
The first important point is that these CuO NPs do not show a real anticancer effect since there is a significant toxicity demonstrated by the first MTT test also in normal cells. The IC50 in normal cells is 83.8 ug/mL while in the two cancer cell lines it just lowers to 53.9 and 58.5 ug/ml. The difference in the IC50 between normal and cancer cells is too low to be described as a specific anticancer effect, furthermore the authors did not perform a statistical analysis supporting that the differences are significant. On the contrary, the cytotoxicity test seems to demonstrate that indeed CuO NP are toxic to mammalian cells in general.
Other than the first experiment showing the results of normal cells, all the other experiments do not compare the investigation of the toxic effects of the CuO NP on the cancer cell lines to the normal cell line, in this way is not possible to establish that the use of the particles is really safe for normal cells, by not triggering the same toxic effects seen in the cancer cell lines. The authors should demonstrate that at the IC50 used for the cancer cells no toxic effects, or minor effects, are seen in the normal cell line, if a real anticancer dose is to be established.
There should be a real discussion of the data, whereas mainly a description of the results is reported, sometimes citing similar results from other studies, but no implications are drawn from the results obtained.
All data presented in the graphs lack a proper statistical analysis and a great deal of information is lacking in the legends of the figures and in the materials and methods section. Just to give some examples:
Fig. 3 There’s no mention of the fluorescent dyes used for the microscopy analysis, what are the green and the red dye staining? What type of microscope? Which Magnification? Besides, at the magnification shown is not possible to clearly observe “morphology changes, such as shrinkage, detachment, membrane blebbing, and deformed shape..” as stated by the authors in the legend.
Fig. 4. A “sub- G0/G1 phase” population is mentioned in the text, but this is not shown in the flow cytometry graph where the other populations are indicated. Again, no statistical analysis here.
Fig. 5a. There’s no quantification of the flow cytometry analysis of the apoptosis Annexin 5 marker vs propidium iodide in the four quadrants. There should be at least a table indicating the percentage of live cells, early apoptotic, late apoptotic, and necrotic, corresponding to the counts of the four quadrants. In some cases, the comment in the text does not match what is seen in the respective quadrant of the picture.
Fig. 6a. The comet assay lacks a quantification in the form of a graph or a table with the proper statistical analysis.
Comments on the Quality of English Language
English must be revised.
Author Response
Dear reviewer,
Greetings.
Thank you for your in-depth comments and detailed questions. The comments given was a way back a great improvement for the research output presented. We have managed address most of the comments to our best and knowledge.
We hope the responses given will be a satisfactory and greatly appreciate for your time given to review our research output.
Thank you.

Round 2
Reviewer 2 Report
Comments and Suggestions for Authors
The artice of Green Synthesis of Copper Oxide Nanoparticles Using Camellia sinensis: Anticancer Potential and Apoptotic Mechanism in HT-29 and MCF-7 Cells is some usefull to read. But there are some mistake to need solve.(1) there is no data about RWPE cells after treated with CuO NPs ;(2) there is no statistical analysis data was conducted in the figure 4b and figure 5b. figure 6b. (3) author should add the detail Mechanisms of cell death;(4) the author should add the postive control in the articles.
Author Response
Comment 1: There is no data about RWPE cells after treated with CuO NPs
Response 1: We thank the reviewer for this observation. Data for the RWPE-1 (normal prostate epithelial) cell line treated with CuO NPs have now been included in the revised manuscript (Figure 4a, explanation in line number: 193-204 and 205-218). These results show the cytotoxic response of normal cells to CuO NPs and allow for a direct comparison with cancer cell lines.
Comment 2: There is no statistical analysis data was conducted in the figure 4b and figure 5b. figure 6b.
Response 2: We appreciate the reviewer’s comment and have addressed this concern in the revised manuscript. Statistical analysis has now been performed for the data presented in Figures 4b, 5b, and 6b. The results are presented as mean ± standard deviation (SD) from at least three independent experiments. Appropriate statistical tests, T-test were applied to assess the significance of differences between treatment groups (line number: 366). The updated figures now include error bars and significance indicators (p < 0.05, p < 0.01, etc.), and the statistical details have been described in the figure legends and the methods section.
Comment 3: Author should add the detail Mechanisms of cell death
Response 3: We thank the reviewer for this valuable suggestion. In the revised manuscript, we have added a detailed explanation of the mechanisms of CuO NP-induced cell death (line number: 503- 520), focusing particularly on ROS-mediated pathways. We believe this enhancement improves the scientific depth and clarity of the manuscript.
Comment 4: The author should add the positive control in the articles.
Response 4: We have now included details regarding the use of commercially available CuO nanoparticles (CuO NPs) as a positive control in our study (results section, line number: 173-175; 190- 191 and methodology section, line number: 604). We also added positive control IC50 graph for commercial CuO NPs (line number: 205). The rationale for selecting CuO NPs is based on their well-documented cytotoxic effects and their ability to generate reactive oxygen species (ROS), which is relevant to the mechanism investigated.

Reviewer 3 Report
Comments and Suggestions for Authors
The authors did not address any of the crucial criticisims raised by the reviewer:
-No statistical analyses (and also no standard deviations in the cytotoxicity test) shown in any of the experiments showing significance.
-No tables showing quantitative analyses and results from the experiments (flow cytometry and comet) were produced by the authors when requested by the reviewer
-No pictures at higher magnification demonstrating the cellular phenomena claimed by the autorhors are shown
-No control of the specific anticancer effect by adding normal cells in all experiments is present.
-No revision of the figure legends and of the materials and methods adding the lacking information is performed, the authors do not even know which fluorophores are used for the live/dead cell assay!
The authors did not modify any substantial part of the manuscript except adding some discussion. The manuscript has not improved in any part as compared to the previous version.
Author Response
Comment 1: No statistical analyses (and also no standard deviations in the cytotoxicity test) shown in any of the experiments showing significance.
Response 1: We have included the statistical analysis and standard deviations for the cytotoxicity study, as well as for other related experiments.
Comment 2: No tables showing quantitative analyses and results from the experiments (flow cytometry and comet) were produced by the authors when requested by the reviewer
Response 2: Thank you for your valuable feedback. We sincerely apologize for the oversight. We are now included the analyses for the comet assay figure 7 (b) i. & ii (line number: 459- 473) and flow cytometry figure 6 (b) I & ii (line number: 310- 338).
Comment 3: No pictures at higher magnification demonstrating the cellular phenomena claimed by the author are shown
Response 3: Thank you for this comment. We have now included the details in Figure 4 (b)- Cell morphological changes observed in comparison to the untreated cells using Nikon TS100 microscope with fluorescence. (scale bar 50 μm; magnification 100x)”.
Comment 4: No control of the specific anticancer effect by adding normal cells in all experiments is present
Response 4:
Thank you for your comment. We acknowledge that the control using normal cells was only included in the preliminary stage of the study, where we conducted the MTT assay on RWPE-1 epithelial cell normal cells. Based on the results, we observed no significant cytotoxicity, which we considered sufficient to indicate the compound’s safety profile. However, we agree that including normal cells in all experiments would strengthen the findings, and we will take this into consideration for future studies.
Comment 5: No revision of the figure legends and of the materials and methods adding the lacking information is performed, the authors do not even know which fluorophores are used for the live/dead cell assay!
Response 5:
We appreciate the reviewer’s comment regarding the need for clarity on the fluorescent dyes used. The live/dead cell assay was performed using Cyto-Dye, a membrane-permeable dye staining viable cells green, and Propidium Iodide (PI), a membrane-impermeant dye labelling non-viable cells red by binding nucleic acids. We have now revised both the Materials and Methods section and figure legends to include this information, which is highlighted in the revised manuscript. (line number: 223- 228)
Comment 6: The authors did not modify any substantial part of the manuscript except adding some discussion. The manuscript has not improved in any part as compared to the previous version.
Response 6:
We sincerely thank the reviewer for their time and constructive feedback. We regret that our previous revision did not sufficiently address the depth of improvements expected. Based on the reviewer’s comments, we have now conducted a thorough, substantive revision of the manuscript, which includes the following major changes:
- Enhanced Figure Legends:
All figure legends have been revised to include essential information such as staining details, fluorescence channels, scale bars, and clearer labelling to improve clarity and reproducibility. - Data Presentation Improvements:
Graphs and figures have been reformatted with appropriate statistical annotations, color coding, and higher-resolution images to improve data visualization and interpretation. - Strengthened Discussion:
The discussion has been substantially expanded to include recent literature comparisons, mechanistic insights, and a critical evaluation of the results within the broader context of the field. - Additional Revisions Throughout the Manuscript:
We carefully restructured and refined the introduction, results, and conclusion sections to improve the logical flow, scientific rigor, and overall readability.We have carefully highlighted all changes in the revised manuscript for the reviewer's convenience.
We deeply appreciate the reviewer’s patience and constructive input, which have greatly contributed to the improvement of this manuscript.

Round 3
Reviewer 2 Report
Comments and Suggestions for Authors
it is ok